# Clinical factors affecting evoked magnetic fields in patients with Parkinson's disease

**Ryoji Naganuma**[1], **Ichiro Yabe**[1]\*, **Megumi Takeuchi**[1], **Kirari Morishita**[2], **Shingo Nakane**[3], **Ryoken Takase**[4,5], **Ikuko Takahashi-Iwata**[1], **Masaaki Matsushima**[1], **Mika Otsuki**[4], **Hideaki Shiraishi**[6], **Hidenao Sasaki**[1]

1 Department of Neurology, Faculty of Medicine and Graduate School of Medicine, Hokkaido University, Sapporo, Hokkaido, Japan, 2 Division of Laboratory and Transfusion Medicine, Hokkaido University Hospital, Sapporo, Hokkaido, Japan, 3 Division of Magnetoencephalography, Hokkaido University Hospital, Sapporo, Hokkaido, Japan, 4 Faculty of Health Sciences/Graduate School of Health Sciences/Department of Health Sciences, School of Medicine, Hokkaido University, Sapporo, Hokkaido, Japan, 5 Department of Radiation Technology, Sapporo City General Hospital, Sapporo, Hokkaido, Japan, 6 Department of Pediatrics, Hokkaido University Hospital, Sapporo, Hokkaido, Japan

\* yabe@med.hokudai.ac.jp

**Data Availability Statement:** All relevant data are within the manuscript and its Supporting Information files.

## Abstract

Studies on evoked responses in Parkinson's disease (PD) may be useful for elucidating the etiology and quantitative evaluation of PD. However, in previous studies, the association between evoked responses and detailed motor symptoms or cognitive functions has not been clear. This study investigated the characteristics of the visual (VEF), auditory (AEF), and somatosensory (SEF) evoked magnetic fields in patients with Parkinson's disease (PD), and the correlations between evoked fields and the patient's clinical characteristics, motor symptoms, and cognitive functions. Twenty patients with PD and 10 healthy controls (HCs) were recruited as participants. We recorded VEF, AEF, and SEF, collected clinical characteristics, performed physical examinations, and administered 10 cognitive tests. We investigated differences in the latencies of the evoked fields between patients with PD and HCs. We also evaluated the correlation of the latencies with motor symptoms and cognitive functioning. There were significant differences between the two groups in 6 of the cognitive tests, all of which suggested mild cognitive impairment in patients with PD. The latencies of the VEF N75m, P100m, N145m, AEF P50m, P100m, and SEF P60m components were greater in the patients with PD than in the HCs. The latencies mainly correlated with medication and motor symptoms, less so with cognitive tests, with some elements of the correlations remaining significant after Bonferroni correction. In conclusion, the latencies of the VEF, AEF, and SEF were greater in PD patients than in HCs and were mainly correlated with medication and motor symptoms rather than cognitive functioning. Findings from this study suggest that evoked fields may reflect basal ganglia functioning and are candidates for assessing motor symptoms or the therapeutic effects of medication in patients with PD.

**Funding:** IY received a grant by Ministry of Health, Welfare, and Labor, Japan (https://www.mhlw.go.jp/english/), there is no grant number. The funders had no role in study design, data collection and analysis, decision to publish, or preparation of the manuscript.

**Competing interests:** The authors have declared that no competing interests exist.

## Introduction

Various cognitive deficits develop during the course of Parkinson's disease (PD). Many types of cognitive deficits can be indicative of neurodegeneration of the cerebral cortex [1]. Evoked responses are electrophysiological recordings that reflect the functioning of sensory pathways in the corresponding part of the cerebral cortex of the stimulated modality, and are widely used in clinical contexts. If cerebral cortex dysfunction can be reflected in evoked responses, these responses may be useful for the quantitative evaluation of PD and in elucidating PD's pathophysiology. Moreover, knowledge of evoked responses in PD may help in the development of novel diagnostic procedures, assessments of medication, and treatment systems for PD. Till date, it is considered that dopamine deficiency affects polysynaptic visual pathway and an increase in the latency of the P100 component of the visual/visually evoked potential (VEP) is reported in PD [2]. In addition, a decrease in the amplitude of the N30 component of the somatosensory evoked potential (SEP) is reported in PD [3]. This could be since he frontal P20-N30-P40 complex (responsible for SEP) is generated with a cortico-subcortico-cortical reentry loop indicating cortical dysfunction in PD. An increase in the latency of wave V (seen as AEP) of the auditory brainstem response has also been reported in PD [4], and it is known that brainstem dysfunction is present in PD patients with dementia. Several clinical characteristics and measures, such as disease duration [5], and scores on the Unified PD Rating Scale (UPDRS) [6–9] and the Mini-Mental State Examination (MMSE) [10–13], have been assessed as possible factors related to these electrophysiological abnormalities, but no consistent associations have been identified.

In this study, we hypothesized that 1) the latencies of the visual (VEF), auditory (AEF), and somatosensory (SEF) evoked magnetic fields would be increased in patients with PD; 2) the latencies would correlate with clinical characteristics, such as disease stage, patients' demographics, or specific motor symptoms and 3) the latencies would correlate with cognitive tests that share the same sensory modality (e.g., auditory tasks and AEF, visual tasks and VEF). We tested these hypotheses by using magnetoencephalography (MEG) as a precursor to consider signal source estimation and brain network analysis in future studies.

## Methods

### Participants

Twenty patients with PD who were admitted to the Department of Neurology, Hokkaido University Hospital, Sapporo, Japan, between July 2017 and March 2019, were recruited for this study. The diagnosis of PD was confirmed according to the MDS clinical diagnostic criteria for Parkinson's disease [14]. Ten age-matched healthy controls (HCs) were also recruited by posters in the hospital and making calls to our acquaintances.

The exclusion criteria for both groups included blindness, deafness, or loss of hand sensation due to sensory organ or brain disorder; an obvious history of brain disease, such as epilepsy, cerebral infarction, or brain surgery; and/or internal metal, such as cardiac pacemakers or deep brain stimulation systems.

The Hokkaido University Hospital Institutional Review Board approved this study. All participants received detailed information regarding their participation, fully understood the explanation, freely agreed to participate, and provided written, informed consent before participation.

### Clinical characteristics and measures

A modified Hoehn–Yahr classification [15] and UPDRS (1987 version) were used to assess the severity of PD in the patients. Physical examinations were conducted by well-trained neurologists, and cognitive tests (described below) were conducted by a clinical psychologist.

All participants underwent a series of 10 tasks to evaluate cognitive functioning. These are accordingly listed in Table 1.

Note that, each task for attention is designed to evaluate a specific attention, but it is also affected by other attention such as working memory, sustained attention and divided attention.

## Medications

For the modified Hoehn–Yahr classification, cognitive tests, UPDRS, and MEG recording, the patients were administered multiple anti-PD drugs, per usual, and the tests were conducted during the on-state, avoiding the off-state. The participants who were inpatients (N = 8) stopped medication for over 12 hours and the UPDRS Part 3 was evaluated again in the off-state for these patients.

**Table 1. List of the cognitive tests.**

| TEST | DESCRIPTION | REFERENCES |
|---|---|---|
| [1]MMSE | a general cognitive test for screening | |
| [2]FAB | a test for frontal lobe functions especially executive function and working memory | [16] |
| noise pareidolia test | a test for visual hallucinations | [17] |
| [3]MoCA-J | a general cognitive test for screening | [18] |
| [4]CAT | a test battery for attention | [19] |
| digit span | short-term memory and working memory | |
| tapping span | visual short-term memory and visual working memory | |
| [5]VCT | visual selective attention | |
| [6]ADT | auditory selective attention | |
| [7]PASAT with 3 s intervals | working memory | |
| [8]TMT A and B | divided attention | [20] |
| [9]WAIS III | a general intelligence test | [21] |
| block design | visual space perception | |
| [10]VPTA | a test battery for visual perception | [22] |
| picture naming | image recognition | |
| symbol recognition character recognition | symbol recognition | |
| famous person naming facial expression recognition | face recognition | |
| line bisection test Albert's test copying (so-called double daisy) | visual space recognition | |
| [11]RCPM | a nonverbal intelligence test | [23] |
| [12]S-PA | a test for verbal paired-associate learning or verbal memory | [24] |

Abbreviations

[1]MMSE: Mini Mental State Examination

[2]FAB: Frontal Assessment Battery

[3]MoCA-J: Japanese version of the Montreal cognitive assessment

[4]CAT: Clinical Assessment for Attention

[5]VCT: visual cancellation task

[6]ADT: auditory detection task

[7]PASAT: paced auditory serial addition task

[8]TMT: Trail Making Test

[9]WAIS: Wechsler Adult Intelligence Scale

[10]VPTA: Visual Perception Test for Agnosia

[11]RCPM: Raven's Colored Progressive Matrices

[12]S-PA: standard verbal paired-associate learning test.

## MEG data collection

MEG data were collected with an Elekta/Neuromag Vectorview 306-channel whole-head neuromagnetometer (Elekta AB, Sweden). In the somatosensory condition, constant current electrical stimulation of 0.2 ms in duration was applied to a unilateral median nerve at 3 Hz and alternated between the right and the left side. Stimulus intensity was set at the supra-motor-threshold. The SEF was recorded 300 times from the whole-head and then averaged.

In the auditory condition, auditory stimuli generated by the STIM program (Compumedics, Abbotsford, VA, Australia) were delivered from a small speaker in a shielded room and guided to a unilateral ear through an air tube. The auditory stimulus was a 2000 Hz tone burst, with a volume of 100 dB sound pressure level at the speaker (99 dB at the ear), a duration of 150 ms, with a 30 ms Hanning window, and was applied to a unilateral ear and alternated between the right and the left side. The AEF was recorded 100 times from the whole-head and then averaged.

In the visual condition, visual stimuli were projected, using an LCD projector outside of the shielded room, onto a screen 45 cm in front of the participant's face, stimulating the unilateral visual hemifield and alternated between the right and the left side. The visual stimulus generated by STIM was a checkerboard pattern with a reversal rate of 2 Hz. The stimulus was a 13 $cm^2$ square that subtended a visual angle of 16.6˚, with an average luminance of 27 $cd/m^2$ and a contrast of 63.0%. The VEF was recorded 300 times from the whole-head and then averaged. Each participant was monitored on a video monitor and alerted when distracted or drowsy.

**Signal processing.**   When describing an evoked field below, the stimulation side is indicated by an upper-case letter in parentheses after the name of the evoked field. In addition, since AEF was recorded from both hemispheres, "c" and "i" are added for the AEF recorded from the contralateral and the ipsilateral hemisphere, respectively, to the stimulation side. For instance, AEF P50m stimulated from the left ear and recorded from the right hemisphere is indicated as AEF P50m (Lc).

A 40 Hz low-pass filter was applied to the VEF data. Similarly, a 30 Hz low-pass and a 4 Hz high-pass filter were applied to the AEF data, and a 100 Hz low-pass and 0.5 Hz high-pass filter were applied to the SEF data using the xplotter program (Neuromag, Helsinki, Finland). After noise processing, the data were exported as.csv files. We calculated the root mean square of the gradiometer pairs within each channel and plotted these as graphs. We quantified the latencies of the evoked fields based on the maximum root mean square peak.

## Statistical analyses

We used JMP ver. 14.0 (SAS Institute, USA) for statistical analyses. When comparing between groups, we used Welch's *t*-test for continuous variables and Wilcoxon signed-rank test for ordinal variables. P-values < 0.05 were considered statistically significant. Descriptive statistics of continuous variables are reported as *mean ± standard deviation* and ordinal variables are reported as *median (interquartile range)*. When evaluating the association between the evoked field responses and clinical characteristics or measures, we used the Pearson correlation coefficient for continuous variables and Spearman's rank correlation coefficient for ordinal variables. Results were considered statistically significant if |R| or $|r_s|$ > 0.6, and p < 0.05. If only one or two scores were obtained on any of the clinical measures that were on an ordinal scale, we excluded them from the statistical analysis. The Bonferroni correction was also performed to evaluate the correlation.

# Results

## Participants and clinical characteristics

There were 13 male and seven female patients with a mean age of 66.9 ± 7.5 years, while the HCs included five men and five women with a mean age was 63.1 ± 5.9 years. There was no

significant difference in age between the two groups. The average disease duration was 12 ± 5.2 years and the median modified Hoehn–Yahr stage was 3 [2.5–3]. Details of the medications that patients were taking are described in S1 Table. The levodopa equivalent dose, calculated based on a report by Tomlinson et al. [25], was 860.2 ± 500.9 mg.

## Clinical measures

In order to assess laterality effects, patients' left and right sides were also designated as the severe side (S) and the mild side (M), based on the UPDRS Part 3 evaluation. Referring to UPDRS Part 3, we compared the total scores of the left upper and lower limbs and that of the right upper and lower limbs. We defined the side with the higher score as S and the side with the lower score as M. If both total scores were equal, we referred to the patient's medical history and defined the side that first developed symptoms as S and the opposite side as M. When comparing patients' S or M with those of HCs, we used the average of the left and right sides in the HCs as the counterparts of the patients' S and M, respectively. The results of the UPDRS are shown in S2 Table. In addition to the usual UPDRS evaluation, eight patients out of the total 20 patients were administered UPDRS part 3 in the off-state since their antiparkinsonian drugs were withdrawn for the purpose of assessing deep brain stimulation (DBS) indications.

A total of 76 scores were obtained from the 10 cognitive tests. The results of the cognitive tests that yielded significant differences between PD and HCs are shown in Table 2. Especially, VCT time of simple symbol, complex symbol and number and ADT hit rate, of the CAT remained significant after Bonferroni correction. For the CAT assessment, two patients were excluded from the analysis because they could not perform the ADT or the paced auditory serial addition task due to not being able to understand the voice on the recording that is used to administer each of those tasks.

## Evoked fields

**Latencies.** We evaluated the responses of each evoked field, distinguishing between the left and right, as well as the S and M, sides. The latencies of the components, and the differences of the latencies between components, of recorded evoked fields that were significantly different between patients with PD and HCs are shown in Table 3. We could not locate AEF P50m (Lc) in two patients and AEF P50m (Ri) in one patient because of an artifact; therefore, they were excluded from the analyses.

The latencies of the VEF N75m (L, R, S, M), P100m (L), N145m (L), AEF P50m (Rc, Li, Si, Mi), and P100m (Rc, Sc, Mc, Li, Ri, Si, Mi) components, and of the SEF P60m (L, R, S, M) component, were significantly increased in patients with PD. Similarly, the latency differences of AEF P100m - 50m (Mc), SEF P60m - N20m (L, R, S, M), and P60m - P35m (L, R, S, M), were increased in patients with PD, and their standard deviations were also increased. Especially, SEF P60m (R, M), P60m –N20m (R, M) and P60m - P35m (R, M) remained significant after Bonferroni correction. No significant left–right difference in latency was seen in either patients with PD or in HCs. Moreover, there was no significant difference in the latency between the S and M sides in patients.

**Site locations.** To assess the sites where the magnetic fields were evoked, we compared the channel number of the sensor from which the maximum root mean square peak of each evoked field was derived, between the patients with PD and the HCs (S3 Table). There were no significant differences between patients with PD and HCs, except for three of the components: VEF N145m (R), AEF P50m (Lc), and SEF N20m (R). Further, after applying Bonferroni correction, the values for these three components became insignificant.

**Table 2. Results of the cognitive tests that were significantly different between patients with Parkinson's disease and healthy control participants.**

| | Patients | | Controls | | p |
|---|---|---|---|---|---|
| **FAB[1]** | 14 | [12.0–15.0] | 17.5 | [16.3–18] | <0.001 |
| Lexical fluency | 2 | [2.0–3.0] | 3 | [3–3] | 0.045 |
| Motor series | 3 | [0.0–3.0] | 3 | [3–3] | 0.025 |
| Go/No-Go | 0.5 | [0.0–3.0] | 3 | [3–3] | 0.043 |
| **MoCA-J[2]** | 23 | [21.8–24.0] | 25.5 | [23.3–27.5] | 0.029 |
| Language [fluency] | 0 | [0.0–1.0] | 1 | [1–1] | 0.044 |
| Abstraction | 1 | [0.0–1.0] | 2 | [1.3–2] | <0.001 |
| Delayed recall | 1.5 | [0.0–2.0] | 3 | [1.5–4.5] | 0.036 |
| **CAT[3]** | | | | | |
| Tapping span (order) | 6 | [5.0–7.0] | 7 | [7–8] | 0.014 |
| VCT[4] simple symbol time (s) | 56.5 | ± 15.3 | 39.8 | ± 8.1 | <0.001* |
| VCT simple symbol hit rate (%) | 99.7 | ± 0.7 | 100.0 | ± 0.0 | 0.042 |
| VCT complex symbol time (s) | 73.7 | ± 24.6 | 44.8 | ± 6.4 | <0.001* |
| VCT complex symbol hit rate (%) | 99.4 | ± 1.3 | 100.0 | ± 0.0 | 0.045 |
| VCT number time (s) | 105.8 | ± 22.7 | 77.9 | ± 12.8 | <0.001* |
| VCT character time (s) | 127.8 | ± 34.7 | 98.2 | ± 17.0 | 0.004 |
| ADT[5] correct answer rate (%) | 95.7 | ± 5.6 | 99.2 | ± 1.0 | 0.018 |
| ADT hit rate (%) | 93.3 | ± 5.5 | 99.2 | ± 1.4 | <0.001* |
| PASAT[6] | 27 | [20.0–41.0] | 57.5 | [54.5–59] | <0.001 |
| **TMT[7]** | | | | | |
| TMT-A (s) | 48.2 | ± 17.9 | 36.5 | ± 11.7 | 0.047 |
| TMT-B (s) | 130.7 | ± 68.0 | 78.4 | ± 25.1 | 0.005 |
| **VPTA[8]** | | | | | |
| Line bisection task score | 1 | [1.0–2.0] | 0 | [0–1] | 0.018 |
| Line bisection task time (s) | 4.7 | ± 3.7 | 2.3 | ± 1.6 | 0.024 |
| Albert's test time (s) | 41.1 | ± 23.8 | 20.7 | ± 5.2 | 0.013 |
| Copying score | 0 | [0.0–1.3] | 0 | [0–0] | 0.025 |
| Copying time (s) | 111.5 | ± 58.8 | 57.7 | ± 28.4 | 0.003 |
| **S-PA[9]** | | | | | |
| No association raw score 1st | 0 | [0.0–2.0] | 2 | [1–5] | 0.016 |
| No association raw score 2nd | 2 | [1.0–4.0] | 5.5 | [4.3–7.8] | 0.004 |
| No association raw score 3rd | 4 | [2.0–6.0] | 7.5 | [4.5–9] | 0.024 |

Data are given as average ± SD or as median [Q1–Q3].

*: Significant after Bonferroni correction. Abbreviations

[1] FAB: Frontal Assessment Battery

[2] MoCA-J: Japanese version of the Montreal cognitive assessment

[3] CAT: Clinical Assessment for Attention

[4] VCT: visual cancellation task

[5] ADT: auditory detection task

[6] PASAT: paced auditory serial addition task

[7] TMT: Trail Making Test

[8] VPTA: Visual Perception Test for Agnosia

[9] S-PA: standard verbal paired-associate learning test.

**Table 3. Results of the latencies of the components of evoked fields that were significantly different between patients with Parkinson's disease and healthy control participants.**

| | Patients | | Controls | | p |
|---|---|---|---|---|---|
| | Average | ± SD | Average | ± SD | |
| **VEF[1] N75m** | | | | | |
| L | 86.7 | ± 8.3 | 80.0 | ± 3.6 | 0.005 |
| R | 88.2 | ± 9.6 | 81.3 | ± 2.6 | 0.007 |
| S[4] | 86.7 | ± 8.3 | 80.6 | ± 2.4 | 0.006 |
| M[5] | 88.1 | ± 9.6 | 80.6 | ± 2.4 | 0.003 |
| **VEF P100m** | | | | | |
| L | 124.1 | ± 17.7 | 111.9 | ± 12.1 | 0.037 |
| **VEF N145m** | | | | | |
| L | 179.3 | ± 20.2 | 165.3 | ± 12.5 | 0.028 |
| **AEF[2] P50m** | | | | | |
| Rc[6] | 56.7 | ± 6.3 | 52.3 | ± 3.1 | 0.015 |
| Li[7] | 67.2 | ± 9.2 | 59.8 | ± 5.3 | 0.009 |
| Si | 64.5 | ± 9.3 | 57.9 | ± 4.6 | 0.015 |
| Mi | 64.7 | ± 11.9 | 57.9 | ± 4.6 | 0.040 |
| **AEF P100m** | | | | | |
| Rc | 100.2 | ± 12.5 | 90.9 | ± 6.7 | 0.012 |
| Sc | 99.2 | ± 13.8 | 91.0 | ± 5.3 | 0.027 |
| Mc | 99.8 | ± 13.8 | 91.0 | ± 5.3 | 0.019 |
| Li | 113.3 | ± 13.2 | 101.1 | ± 6.2 | 0.002 |
| Ri | 108.0 | ± 12.9 | 99.4 | ± 6.2 | 0.020 |
| Si | 109.4 | ± 13.6 | 100.2 | ± 4.7 | 0.012 |
| Mi | 111.8 | ± 12.9 | 100.2 | ± 4.7 | 0.001 |
| **AEF P100m - P50m** | | | | | |
| Mc | 43.9 | ± 8.9 | 38.0 | ± 5.1 | 0.029 |
| **SEF[3] P60m** | | | | | |
| L | 67.0 | ± 12.7 | 53.4 | ± 13.0 | 0.01 |
| R | 67.1 | ± 12.0 | 48.1 | ± 9.2 | <0.01* |
| S | 63.6 | ± 12.3 | 50.8 | ± 9.3 | <0.01 |
| M | 70.5 | ± 11.3 | 50.8 | ± 9.3 | <0.01* |
| **SEF P60m - N20m** | | | | | |
| L | 43.5 | ± 11.6 | 30.7 | ± 12.5 | 0.014 |
| R | 43.9 | ± 11.8 | 25.7 | ± 8.9 | <0.001* |
| S | 40.4 | ± 11.9 | 28.2 | ± 9.1 | 0.005 |
| M | 47.0 | ± 10.5 | 26.9 | ± 8.5 | <0.001* |
| **SEF P60m - P35m** | | | | | |
| L | 36.8 | ± 12.4 | 23.5 | ± 12.8 | 0.014 |
| R | 36.7 | ± 11.9 | 18.7 | ± 8.9 | <0.001* |
| S | 33.3 | ± 11.8 | 21.1 | ± 9.2 | 0.005 |
| M | 40.2 | ± 11.4 | 19.9 | ± 8.5 | <0.001* |

Latencies are reported in ms.

*: Significant after Bonferroni correction. Abbreviations

[1]VEF: visual evoked magnetic field

[2]AEF: auditory evoked magnetic field

[3]SEF: somatosensory evoked magnetic field

[4]S: severe side stimulation

[5]M: mild side stimulation

[6]c: contralateral side recording

[7]i: ipsilateral side recording.

### Correlation between evoked fields and clinical characteristics and measures

Table 4 shows where there were significant correlations between the scores of particular clinical characteristics or measures, and the latencies, or latency differences, of components of evoked fields, in patients with PD. The correlation coefficients are reported in S4 Table. With respect to medications, we included L-DOPA, dopamine agonists, and the levodopa equivalent dose, except for pramipexole and pergolide, which were taken by a small number of patients. The clinical scores and evoked fields that were assessed here were significantly different between patients with PD and HCs or were assessed only in PD patients.

Multiple medications and motor symptoms correlated with latencies or latency differences, but the CAT was the only cognitive test for which the scores correlated with the evoked field data. After performing a Bonferroni correction, only three pairs of latencies, or latency differences, showed statistically significant correlations with clinical scores or medication: levodopa/benserazide and AEF P100m (Ri, Mi) as well as UPDRS Part 3 Speech in the on-state and AEF P100m (Ri). A similar analysis was performed for the HCs, but no correlations were shared between HCs and patients with PD.

## Discussion

### Cognitive functioning

There were significant differences in MoCA-J (total score, Language [fluency], abstraction, delayed recall), S-PA (no association raw score 1st, 2nd, 3rd), FAB (Lexical fluency, Motor series, Go/No-Go), TMT-A, B, CAT (tapping span [order], VCT [simple symbol time, simple symbol hit rate, complex symbol time, complex symbol hit rate, number time, character time], ADT [correct answer rate, hit rate], PASAT), and VPTA (line bisection task [score, time], Albert's test time, copying [score, time]) scores between patients with PD and HCs, all of which reflected a decrease in cognitive functioning in PD patients.

FAB is a test that assesses frontal lobe functions involving executive dysfunction and working memory, through tasks such as lexical fluency, motor series, and Go/No-Go. A previous study showed that the FAB total score was low in patients with PD [26]. In addition, a study that used the Wisconsin card sorting test showed frontal lobe dysfunction, especially executive dysfunction, in patients with PD [27]. Both studies are consistent with the findings of this study. Ohta and Suzuki [28] have reported that PD patients with mild cognitive impairment had lower scores in MoCA-J than in MMSE, especially on Short-Term Memory, Language (fluency), and Delayed recall. This is almost the same as our study because our patients had similar MMSE scores but lower scores in MoCA-J (Language (fluency), and Delayed recall) than HCs. The TMT-A, B, and CAT items, including tapping span, VCT, ADT, and PASAT, are all affected by deficits in sustained attention. Maruyama [29] reported sustained attention deficits in PD, and the results of this study could also be related to sustained attention deficits. Sustained attention deficits are likely to be present in PD since among all the cognitive tests performed in our study, VCT and ADT remained significant after Bonferroni correction.

A previous report suggested that visual perception was impaired in patients with PD [26]. While the tasks in that study were different from those in this study, significantly higher VPTA scores in patients with PD in this study may also be due to visual impairment. It may also be that the scores of items with actions such as TMT, CAT (tapping span, VCT, ADT, PASAT), and VPTA (line bisection task, Albert's test, copying) were significantly increased in PD patients due to bradykinesia. There is no previous report on S-PA in patients with PD. S-PA is a test on verbal paired-associate learning, which is evaluated qualitatively based on raw scores and obtaining rating scores. Although the rating score was not significantly different between

**Table 4. Results of significant correlations between the latencies of components of evoked fields and scores of clinical characteristics or measures in patients.**

| | | VEF[1] | VEF | VEF | AEF[2] | AEF | AEF | SEF[3] | SEF | SEF |
|---|---|---|---|---|---|---|---|---|---|---|
| | | N75m | P100m | N145m | P50m | P100m | P100m - P50m | P60m | P60m - N20m | P60m - P35m |
| **CAT[4]** | | | | | | | | | | |
| ADT[5] | hit rate | | | | | | | L, M | L, M | L, M |
| | correct answer rate | | | | | | | | | S |
| VCT[6] (character) time | | | L | | | | | | | |
| **Medication** | | | | | | | | | | |
| Levodopa/benserazide | | M | | | | Rc[12], Mc, Li[13], Ri*, Si, Mi* | Mc | | | |
| Rotigotine | | | | | Rc | | | | | |
| Ropinirole | | L | | | | | | | | |
| **UPDRS[7] Part 2 on** | | | | | | | | | | |
| Total score | | | L | | | | | | | |
| Speech | | | | | | | | R, S | R, S | R, S |
| Swallowing | | | L | | | | | | | |
| **UPDRS Part 3 on** | | | | | | | | | | |
| Speech | | | | | | Ri* | | | | |
| Gait | | | L | | | | | | | |
| **UPDRS Part 3 off** | | | | | | | | | | |
| Total | | | | | | Mi | | | | |
| Speech | | L, M | | | | | | | | |
| Facial expression | | | | | Li, Mi | Rc, Sc, Mc, Li | | R, S | S | S |
| Rigidity | RLE[8] | | | | | | | R | | |
| | LLE | R | | | | | | R | | |
| | S[9]LE | | | | | Mi | | S | S | S |
| | M[10]LE | | | | | Mi | | | | |
| Tremor at rest | LUE[11] | | L | | | | | | | |
| | RUE | | L | | | | | | | |
| | SLE | | M | | | | | | | |
| Action or postural tremor | L | | L | L | | | | | | |
| | R | | L | L | | | | | | |
| Finger taps | R | | L | L | | | | | | |
| | S | | M | | | | | | | |
| | M | | M | | | | | | | |
| Hand movements | S | | M | | | | | | | |
| | M | | M | | | | | | | |
| Rapid alternating movements | L | | | | | Li | | | | |
| | S | | | | | Si | | | | |
| | M | | | | Mi | Sc | | | | |
| Leg agility | L | | R | | | | | | | |
| | R | | R | | | | | | | |
| | S | | | | | Mi | | | | |
| | M | | | | | Mi | | | | |
| Arising from chair | | | | | | Li | | R, S | S | |
| Posture | | | L | L | | Li | | R, S | S | S |
| Gait | | | | | Li, Mi | Mc, Li, Si | | R, S | S | S |
| Postural stability | | | | | | Rc, Sc, Li, Mi | | R, S | S | |
| Body bradykinesia | | R | | | | Li, Mi | | R, S | S | S |

*(Continued)*

**Table 4.** (Continued)

| | VEF[1] | VEF | VEF | AEF[2] | AEF | AEF | SEF[3] | SEF | SEF |
|---|---|---|---|---|---|---|---|---|---|
| | N75m | P100m | N145m | P50m | P100m | P100m - P50m | P60m | P60m - N20m | P60m - P35m |
| **UPDRS off Total score** | | | | | | | | | |
| | | | | Li, Mi | Li, Mi | | R, S | S | S |

Orange cell: positive correlation, Blue cell: negative correlation

*: Significant after Bonferroni correction. Abbreviations

[1]VEF: visual evoked magnetic field

[2]AEF: auditory evoked magnetic field

[3]SEF: somatosensory evoked magnetic field

[4]CAT: Clinical Assessment for Attention

[5]ADT: auditory detection task

[6]VCT: visual cancellation task

[7]UPDRS: Unified Parkinson's Disease Rating Scale

[8]LE: lower extremity

[9]S: severe side

[10]M: mild side

[11]UE: upper extremity

[12]c: contralateral side recording

[13]i: ipsilateral side recording.

patients with PD and HCs, PD patients may have verbal memory impairment because the raw scores in all three trials were significantly low. Taken together, these results suggest that PD patients have at least mild cognitive impairment from PD.

## Evoked fields

The latencies of VEF, AEF, and SEF were increased in PD patients and they also had large SDs. This indicates that evoked field latencies in PD patients were quite variable, possibly providing indirect evidence that the pathophysiology of PD affects evoked field latencies. To the best of our knowledge, no study has been conducted on the association between evoked responses and cognitive functioning, except for a study assessing MMSE [10–13]. Thus, this study provides new insights into the association between evoked responses and cognitive functioning in patients with PD.

With respect to the clinical measures, CAT was the only cognitive test that was significantly associated with evoked fields; however, numerous elements of the UPDRS and patients' medications were associated with evoked fields. This suggests that medication and motor symptoms, rather than cognitive function, influence the latencies of evoked fields. When a Bonferroni correction was performed, only three pairs of associations remained significant: levodopa/benserazide and AEF P100m (Ri, Mi) as well as UPDRS Part 3 Speech in the on-state and AEF P100m (Ri). These results indicate that the evoked fields are influenced mainly by medication and motor symptoms. Because basal ganglia play a major role in motor symptoms of PD and their function is regulated by dopamine, it may be that evoked fields reflect basal ganglia function.

On the other hand, there were no significant correlations that survived Bonferroni correction between UPDRS Part 3 in the off-state and latencies of the evoked fields, even though there were several significant correlations before the correction. There may be various reasons for this finding. First, since MEG was conducted during the on-state, evoked fields may simply

reflect the motor symptoms in the on-state. Second, this result may indicate that motor symptoms do not affect evoked fields. Third, this result may be erroneous because UPDRS Part 3 in the off-state was assessed in only a small number of patients. Nevertheless, elements of the UPDRS Part 3 in the off-state are factors that could plausibly influence the evoked brain magnetic field; thus, these findings should be assessed in a larger number of subjects in future studies.

## VEF

When discussing the results of this study, we also refer to reports on VEP because the N75m, P100m, and N145m VEF components correspond to the N75, P100, and N145 VEP components, respectively [30]. Several studies have reported that the latency of the VEP P100 component is increased in patients with PD [13, 31], while one study reported that the latencies of the VEF N75m and P100m components were increased in patients with PD [8]. Fujisawa et al. [8] also reported that the P100m –N75m latency difference was increased in patients with PD, but this latency difference was not increased in the present study. No previous study has reported increased latencies of VEP N145 or VEF N145m components in patients with PD.

The N75m and P100m components are thought to originate from the visual cortex, and the N145m component is thought to originate from the extrastriate cortex [32]. In this study, the latencies of the N75m (L, R, S, M), P100m (L), and N145m (L) components were increased in PD, but central conduction times (P100m - N75m, N145m - P100m, and N145m - N75m latency differences) did not differ between patients with PD and HCs. These results support a previous study that reported that the origin of the conduction delay may be peripheral rather than cortical [8]. Processing delays in the retina or along the pathway to the cortex may contribute more than that of the cortex to the delay in the VEF. It should be noted that where there were significant differences between PD patients and HCs in the latencies of particular components (i.e. N75m [L, R, S, M], P100m [L], and N145m [L]), these differences were not due to variations in location, as there were no significant differences between patients with PD and HCs in the site locations of these components.

For cognitive tests, CAT VCT (characters) time was positively associated with the latency of the P100m (L) component. VCT reflects visual selective attention; VCT time is prolonged in individuals with a visual selective attention deficit. Thus, based on our results, the P100m (L) latency may reflect visual selective attention. VCT time is also affected by bradykinesia of Parkinsonism because it partly depends on writing speed, but there was no association between the latency of the P100m (L) component and the UPDRS Part 3 body bradykinesia score. This helps support our hypothesis that the P100m (L) may reflect visual selective attention. Okuda et al. [11] reported that the P100 latencies were negatively correlated with the MMSE score in patients with PD, but no such association was found in this study.

In terms of motor symptoms, both the total score and the swallowing score of UPDRS Part 2 in the on-state, and the Walk score of UPDRS Part 3 in the on-state, were positively associated with latency of the P100m (L) component. These results were consistent with those of previous studies that found that the latency of the P100 VEP component in patients with PD was positively correlated with the Hoehn–Yahr stage as well as disease severity [2, 6, 33]. It remains unclear why the latency of the N145m component, which is evoked after the P100m, had no significant association with any motor symptom associated with the latency of P100m, but it may be because the SDs of the latencies of the N145m component were greater than those of the P100m, in both PD patients and controls.

For the UPDRS Part 3 in the off-state, action or postural tremor (L, R) as well as posture scores, were positively associated with the latency of the P100m (L) and the N145m (L)

components, while tremor at rest (R upper extremity [UE], LUE) were positively associated with the latency of the N75m (L) component. However, there were negative associations between the finger taps (R) score and the latency of the P100m (L) and N145m (L) components; speech score and the latency of the N75m (L) component; speech, tremor at rest (S lower extremity [LE]), finger taps (S, M), and hand movement (S, M) scores, and the latency of the N75m (M) component; rigidity (LLE), leg agility (L, R), and body bradykinesia scores, and the latency of the N75m (R) component. The latencies of the P100m (L) and N145m (L) components were negatively correlated with finger taps (L), while they were positively correlated with other motor symptoms, posing a contradiction. This may have been due to the small number of patients that were assessed for the UPDRS Part 3 in the off-state.

In terms of medication, levodopa/benserazide was positively associated with the latency of the N75m (M) component, and ropinirole was positively associated with the latency of the N75m (L) component, which seems to reflect the need for more medication as motor symptoms progress. There have been no previous reports of the association between daily dosage of anti-parkinsonian drugs and VEF. However, it has been reported that the increased latency of the P100 VEP component in patients with PD is improved by L-DOPA treatment [3, 6]. It may be that increased VEF latencies, latency differences, and correlations between latencies and clinical characteristics and measures, would be more pronounced and detectable if the evoked fields were obtained without medication. Nevertheless, it should be noted that when a Bonferroni correction was performed, no association between latencies of the VEF and scores on the clinical measures remained statistically significant.

### AEF

To the best of our knowledge, there have been no reports on increases in the latencies of the AEF P50m and P100m components. As it has been reported that interhemispheric latency differences in AEF P50m and P100m were significantly increased in PD patients in the left-ear condition [34], we calculated the interhemispheric latency differences in the left-ear condition in this study: P50m (Li)—P50m (Lc) [PD 10.3 ± 7.2 vs. HCs 8.5 ± 7.8, p = 0.25] and P100m (Li)—P100m (Lc) [14.5 ± 9.0 vs. 12.6 ± 7.3, p = 0.33]. No significant differences were found.

The AEF P50m component is thought to be equivalent to the Pb component of the middle latency response (MLR) [35]; the AEF P100m component is thought to be equivalent to the N1 component of the auditory late response (ALR) [36], and both of these components (the Pb and the N1) originate from the cerebral cortex. It has previously been reported that the amplitude of the middle latency response of the Pb component is increased in PD patients [37], but there have been no reports of increased latencies of the middle latency response of the Pb component or of the auditory late response of the N1 component in PD patients.

In this study, the latencies of the AEF P50m (Rc, Li, Si, Mi) and P100m (Rc, Sc, Mc, Li, Ri, Si, Mi) components were increased. Since the latency of wave V of the auditory brainstem response has been reported to be increased in PD patients [4], latencies of the AEF P50m and P100m components in PD patients may be more increased in the periphery than in the thalamus. On the other hand, the latency of the AEF P100m - P50m (Mc) difference increased, and the delay was more apparent in the P100m than in the P50m component; therefore, the latencies may also be increased at the cortex. Note that because the sites where AEF P50m (Rc, Li, Si, Mi) and P100m (Rc, Sc, Mc, Li, Ri, Si, Mi) components were evoked did not differ between patients with PD and HCs, the observed differences were not related to locational differences.

While there was no association between cognitive tests and AEF, in terms of motor symptoms, the speech score of the UPDRS Part 3 in the on-state was positively associated with the latency of the P100m (Ri) component. This association remained significant when a

Bonferroni correction was performed. Since the language area is generally located in the left temporal lobe, the association between the AEF recorded from the right temporal lobe and language function is unknown; it may not be related to language function, but rather to other factors that affect articulation (such as voice volume).

For other motor symptoms, there were numerous associations between elements of the UPDRS Part 3 in the off-state and the AEF latencies, but none of these were statistically significant after Bonferroni correction. For instance, the rapid alternating movements (M) score was positively associated with the latency of the P50m (Mi) component, and the rapid alternating movements (L, S) score was positively associated with the latency of the P100m (Li, Si) component; these correlations coincided with the laterality of the examined side and the recording side of the evoked field. These results suggest that AEF may be associated with motor symptoms in the ipsilateral hand. Since rising from a chair, posture, gait, and postural stability scores were all positively associated with the latency of the P100m (Li) component, an increased latency in the P100m (Li) component may reflect a postural reflex disorder. Rigidity (SLE, MLE), leg agility (S, M), and body bradykinesia scores, which are all related to rigidity of the legs, were all positively associated with the latency of the P100m (Mi) component. Thus, an increased latency of the P100m (Mi) component may reflect leg rigidity.

The total score of the UPDRS in the off-state, which included Part 1, Part 2 in the off-state, Part 3 in the off-state, and Part 4, was positively associated with the latencies of the P50m (Li, Mi) and P100m (Li, Mi) components. This may be partly due to the score for Part 3 in the off-state driving the effect. However, the number of elements of Part 3 in the off-state that correlated with the latencies of the P50m (Li, Mi) and P100m (Li, Mi) components was too small for the number of elements of the total score of the UPDRS in the off-state. It is possible that Part 1, Part 2 in the off-state, and Part 4 themselves were potentially correlated with the latencies of the P50m (Li, Mi) and P100m (Li, Mi) components.

A previous study reported that the latency of wave V of the auditory brainstem response was increased in PD patients, but was not associated with motor symptoms or disease severity [4], and there have been no previous studies reporting associations of the AEF, MLR, or ALR with motor symptoms or disease severity. Since this is the first study of an association between AEF and motor symptoms, which are causally related, further accumulation of data is required in the future.

A positive association was found between the daily dosage of levodopa/benserazide and the latency of the P100m (Rc, Mc, Li, Ri, Si, Mi) component and the latency difference of P100m - P50m (Mc). The association with the latency of the P100m (Ri, Mi) component remained significant after Bonferroni correction. Additionally, an association was found between rotigotine and the latency of the P50m (Rc) component. Since the latency of the P100m (Ri) component had a strong positive association with the speech score of the UPDRS Part 3 in the on-state, and the latencies of the P100m (Li, Mi) component were also positively associated with the UPDRS Part 3 score in the off-state, these associations with medication may reflect PD severity.

It has been reported that dopamine administration affects the auditory responses in the inferior colliculus in mice [38]. In addition, there have been reports of rats having increased auditory-evoked responses from the brainstem to the basal ganglia after dopamine loading [39]. Dopamine neurons are associated with the auditory tract, and auditory evoked responses may reflect the state of dopamine neurons as affected by medication or pathophysiology.

## SEF

Since the SEF N20m component originates from area 3b of the primary sensory cortex, which is also the origin of the SEP N20 component [40, 41], these two are considered equivalent.

However, since the P35m and P60m SEF components also originate from area 3b of the primary sensory cortex [40] and the N30 SEP component originates from the supplementary motor area [42], they are considered to be different.

Most previous studies on somatosensory evoked responses found that the amplitude of the N30 SEP component decreased in PD subjects. Most of the few existing reports stated that N20 SEP component does not differ in patients with PD, but one report stated that the amplitude of the N20m SEF component on the symptomatically more severe side in PD patients is greater than that on the contralateral side [43]. In this study, we did not evaluate the amplitude of the N20m component.

In this study, the latency of the P35m component was not significantly different between patients with PD and HCs, but the latency of the P60m component was significantly higher in PD patients. Furthermore, for central conduction time, both P60m - P35m and P60m - N20m latency differences were significantly increased in the patients with PD, regardless of the stimulating side. This implies that only the latency of the P60m component increased. To the best of our knowledge, an increase in latency of the P60m component in PD has not been reported previously. However, as SEF P60m (R, M), SEF P60m –P35m (R, M) and SEF P60m –N20m (R, M) values remained significant after Bonferroni correction among all the evoked fields, it is strongly suspected that P60m is affected by underlying PD pathology. Since the sites where the P60m (L, R, S, M) component were evoked were not different between patients with PD and HCs, the delay in the latency of the P60m component in PD patients was not due to location differences. In addition, the N20m, P35m, and P60m components originate from the same area of the primary sensory cortex (area 3b), and the latencies of the N20m and P35m components did not differ between patients with PD and HCs. As neurodegeneration of the primary sensory cortex rarely occurs in PD patients, it is thought that the delay of the P60m component does not occur in the sensory cortex, but rather in the pathway after P35m.

For cognitive tests, the CAT ADT hit rate was negatively associated with the latency of the P60m (L, M) component and the latency differences of P60m –N20m (L, M) and P60m –P35m (L, M). The CAT ADT correct-answer rate was also negatively associated with the P60m - P35m (S) latency difference. ADT assesses selective auditory attention, and hit rates and correct-answer rates decrease as selective auditory attention is disturbed. Hence, the latency of the P60m component may be associated with selective auditory attention, even though no association was found between ADT and somatosensory processing or parietal lobe activity.

For motor symptoms, numerous associations were found between UPDRS scores and SEF latencies. As indicated above, when the latency difference from the P60m component is considered to be equal to the latency of the P60m component, the results can be restated as follows: the scores for speech in UPDRS Part 2 in the on-state; facial expression, rising from chair, posture, gait, postural stability, and body bradykinesia in UPDRS Part 3 in the off-state; and the UPDRS total score in the off-state, were all positively associated with the latency of the P60m (R, S) component. Scores for rigidity (RLE, LLE) in UPDRS Part 3 in the off-state was positively associated with the latency of the P60m (R) component, and scores for rigidity (SLE) in UPDRS Part 3 in the off-state was positively associated with the latency of the P60m (S) component. The latency of the P60m (R) and/or P60m (S) component had numerous positive associations with UPDRS scores; hence, the latency of the P60m (R, S) component may reflect motor symptoms of PD. Previous studies have reported that the amplitude of the N30 SEP component is decreased in patients with PD and is negatively associated with motor symptoms [3, 44–46]. It has also been suggested that SEPs from the frontal scalp sites could be considered markers of the functionality of a cortico-subcortico-cortical loop [47]. However, we could not find any report on P60m SEF latency. Considering that in this study, only the latency of the P60m component differed between patients with PD and HCs, and was associated with motor

symptoms, the latency of the P60m component may reflect basal ganglia function. Nevertheless, it should be noted that when a Bonferroni correction was performed, no association between SEF and UPDRS scores remained significant. For medication, we found no significant association with SEF latencies.

## Association between evoked fields from different sensory stimuli

For the evoked fields that showed a significant difference between PD patients and HCs, we compared two pairs of evoked fields of different sensory stimulations in PD patients. For the SEF, latency differences (P60m –N20m, P60m –P35m) were considered equivalent to the P60m of ipsilateral stimulation and were excluded from analysis. Four pairs demonstrated significant associations: latencies of the SEF P60m (S) component and the AEF P100m (Mc) component ($R = 0.66$, $p = 0.001$), latency of the SEF P60m (S) component and the latency difference of the AEF P100m - P50m (Mc) ($R = 0.62$, $p < 0.003$), latencies of the SEF P60m (R) component and the AEF P100m (Mc) component ($R = 0.69$, $p < 0.001$), and the latency of the SEF P60m (R) component and the latency difference of AEF P100m –P50m (Mc) ($R = 0.71$, $p < 0.001$). In addition, the latencies of the SEF P60m (R, S) and AEF P100m (Mc) components both showed positive associations with the scores for facial expression and gait in UPDRS Part 3 in the off-state. Based on these results, the delays may reflect basal ganglia function and may occur in parallel with the progression of motor symptoms. However, the latencies of the SEF P60m (R, S) component and the latency difference of AEF P100m –P50m (Mc) had no common associations. Furthermore, these four associations became nonsignificant after Bonferroni correction ($p = 0.212$, $0.478$, $0.101$, and $0.067$, respectively).

## Limitations

There were some limitations to this study. First, MEG was performed on patients in the on-state, and thus, the results may have been affected by medication. In addition, for UPDRS Part 3, we may have identified false-positive associations because only eight patients were evaluated in the off-state. In the future, it will be necessary to assess UPDRS, cognitive functions, and MEG recordings in the off-state for all participants.

Second, there were few associations between the cognitive assessments and evoked fields, considering the number of cognitive tests that were administered. This may be because the VEF, AEF, and SEF are reflective of processing in the primary sensory cortex, while cognitive tasks are processed at higher levels of the sensory cortex. The association between event-related potentials, which are thought to reflect higher cognitive function, and cognitive functioning should be assessed. In addition, the cognitive impairment of the patients in this study was mild. Since abnormal evoked potentials are more likely to be observed in patients with dementia [11–13, 33, 48, 49], this population bias may underlie the few significant associations found. Alternatively, more participants needed to be included because our patient population was heterogeneous, and we examined numerous parameters.

Third, the number of HCs was small. It would have been ideal if the number of HCs was the same as the number of patients, but it was difficult since we could not recruit more controls. This was due to our inability to find new HC's as the study's approval had also expired. However, the results were reliable for the number chosen owing to the homogeneity of the control group in terms of the SD's and interquartile range despite the controls being recruited randomly. A confirmation study to verify the same could be conducted at a later point in time.

Lastly, when assessing the site of sensors, the actual inter-sensor distance of MEG was not necessarily ordered by the channel number; in the future, the actual two-dimensional distribution rather than the rank scale should be used.

## Conclusions

We investigated the characteristics of cognitive functioning and evoked fields in patients with PD. PD patients presented with cognitive impairment on assessment tests such as MoCA-J, S-PA, FAB, TMT-A, B, CAT, ADT and VPTA. Especially, sustained attention deficits are likely to be present in PD. The latencies of the VEF N75m, P100m, N145m, AEF P50m, P100m, and SEF P60m components were greater in PD patients than in HCs. The increased latencies of VEF, AEF, and SEF were correlated mainly with medication and motor symptoms and less with cognitive tasks. Interestingly, AEF P100m and SEF P60m latency could strongly reflect basal ganglia functioning as per our findings. These specific responses are candidates for assessing motor symptoms or drug treatment effects in PD patients.

## Supporting information

**S1 Table. Details of the medication taken by patients.** Abbreviations: [1]COMT, catechol-*O*-methyltransferase; [2]LED, levodopa equivalent dose.
(DOCX)

**S2 Table. Scores on the Unified Parkinson's Disease Rating Scale (UPDRS) by patients with Parkinson's disease.** Data are given as median [Q1–Q3]. Abbreviations: [1]UE: upper extremity, [2]LE: lower extremity.
(DOCX)

**S3 Table. Differences in individual site locations between patients with Parkinson's disease and healthy control participants.** A comparison of the channel numbers of the sensor from which the maximum root mean square peak of each evoked field was derived. Each channel was subjected to the Wilcoxon signed-rank test as an order variable. Abbreviations: [1]VEF: visual evoked magnetic field, [2]AEF: auditory evoked magnetic field, [3]SEF: somatosensory evoked field, [4]p: p value.
(DOCX)

**S4 Table. Correlation coefficients between clinical characteristics and measures, and evoked fields.** Abbreviations: [1]CAT: Clinical Assessment for Attention, [2]ADT: auditory detection task, [3]SEF: somatosensory evoked magnetic field, [4]M: mild side stimulation, [5]S: severe side stimulation, [6]VCT: visual cancellation task, [7]VEF: visual evoked magnetic field, [8]AEF: auditory evoked magnetic field, [9]i: ipsilateral side recording, [10]c: contralateral side recording, [11]UPDRS: Unified Parkinson's Disease Rating Scale, [12]LE: lower extremity, [13]UE: upper extremity.
(DOCX)

## Acknowledgments

This work was supported in part by a Grant-in-Aid for the Research Committee of CNS Degenerative Diseases under Research on Measures for Intractable Diseases from the Ministry of Health, Welfare, and Labour, Japan. We thank Naoki Nishimoto, Ph.D., for advice regarding statistical analysis, Noriki Ochi for conducting MEG evaluations, Atsushi Shimojo for advice of MEG analysis, and Editage (www.editage.com) for English language editing.

## Author Contributions

**Conceptualization:** Ryoji Naganuma, Kirari Morishita, Hideaki Shiraishi, Hidenao Sasaki.

**Data curation:** Ichiro Yabe.

**Formal analysis:** Ryoji Naganuma.

**Funding acquisition:** Ichiro Yabe.

**Investigation:** Ryoji Naganuma, Megumi Takeuchi, Kirari Morishita, Shingo Nakane, Ryoken Takase.

**Methodology:** Ichiro Yabe, Mika Otsuki, Hideaki Shiraishi, Hidenao Sasaki.

**Project administration:** Ichiro Yabe, Hidenao Sasaki.

**Supervision:** Ikuko Takahashi-Iwata, Masaaki Matsushima, Mika Otsuki, Hideaki Shiraishi.

**Writing – original draft:** Ryoji Naganuma.

**Writing – review & editing:** Ichiro Yabe, Ikuko Takahashi-Iwata, Masaaki Matsushima.

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
