## [Decision Letter · Decision Letter 0]

8 Jun 2020

PONE-D-20-11314

Clinical factors affecting evoked magnetic fields in patients with Parkinson's disease

PLOS ONE

Dear Dr. Yabe,

Thank you for submitting your manuscript to PLOS ONE. After careful consideration, we feel that it has merit but does not fully meet PLOS ONE’s publication criteria as it currently stands. Therefore, we invite you to submit a revised version of the manuscript that addresses the points raised during the review process.

We look forward to receiving your revised manuscript.

Kind regards,

Jiaojian Wang

Academic Editor

PLOS ONE

Journal Requirements:

Additional Editor Comments (if provided):

We have recieved two reviewers' comments. One suggests major revision, and the other recommends acceptance. Thus, a minor revision is needed for this submission.

Reviewers' comments:

Reviewer's Responses to Questions

**Comments to the Author**

1. Is the manuscript technically sound, and do the data support the conclusions?

Reviewer #1: Partly

Reviewer #2: Yes

2. Has the statistical analysis been performed appropriately and rigorously? 

Reviewer #1: No

Reviewer #2: Yes

3. Have the authors made all data underlying the findings in their manuscript fully available?

Reviewer #1: No

Reviewer #2: Yes

4. Is the manuscript presented in an intelligible fashion and written in standard English?

Reviewer #1: No

Reviewer #2: Yes

5. Review Comments to the Author

Reviewer #1: The paper compared multiple cognitive functions between PD patients and controls using various cognitive tests and explored the relationship between cognitive performances and evoked magnetic fields. Although sever significant results were described, there are several problems about sample, statistical analysis and results interpretation. Specific concerns are described below:

1. There is too small sample size, especially the HC group. I strongly suggest to recruit more healthy participants and re-analyzed again.

2. In Introduction section, more special description about the different evoked potential, what specific evoked potential reflect physiological feature? It is wonderful if hypothesis was present based on physiological feature.

3. In Clinical characteristics and measures, cognitive function should be provided for all tasks and its sub-domains.

4. There are patients in the off-state and on-state, what difference between two kinds of patients.

5. Multiple corrections should also be conducted for group comparison.

6. In Conclusion section, authors stated that evoked fields reflect basal ganglia function, what cognitive function or evoked fields are linked with ganglia function based on your results or previous studies?

Reviewer #2: In this manuscript, the authors investigated the characteristics of the visual (VEF), auditory (AEF), and somatosensory (SEF) evoked magnetic fields in patients with Parkinson’s disease (PD), as well as the correlations between evoked fields and the patient’s clinical characteristics, motor symptoms, and cognitive functions. The authors found that the latencies of the VEF N75m, P100m, N145m, AEF P50m, P100m, and SEF P60m components increased significantly in patients with PD. Besides, the latencies were found to be correlated with medication and motor symptoms. The reported results are novel and provide new insights into the understanding of the pathophysiology of PD. The manuscript is well organized,technically sound and well-written.

6. PLOS authors have the option to publish the peer review history of their article (what does this mean?). If published, this will include your full peer review and any attached files.

Reviewer #1: No

Reviewer #2: No

---

## [Author Response · Author response to Decision Letter 0]

30 Jun 2020

PONE-D-20-11314

Clinical factors affecting evoked magnetic fields in patients with Parkinson's disease

Dear Editor and Reviewers:

Thank you for your e-mail on June 8, 2020 informing us that you would be willing to consider a revised version of our manuscript entitled “Clinical factors affecting evoked magnetic fields in patients with Parkinson's disease”. We found the points raised by the reviewers to be most helpful, and we have revised our manuscript accordingly. Please find attached the revised manuscript and our detailed point-by-point responses to the comments of the reviewers. We hope that our paper is now acceptable for publication in PLOS ONE, and we thank you for your consideration.

Sincerely yours,

Ryoji Naganuma, Ichiro Yabe

Address: Department of Neurology, Faculty of Medicine and Graduate School of Medicine, Hokkaido University, Kita 15, Nishi 7, Kita-ku, Sapporo 060-8638, Japan

Telephone: 81-11-706-6028

Fax; 81-11-700-5356

E-mail: yabe@med.hokudai.ac.jp

>Reviewer #1

> 1. There is too small sample size, especially the HC group. I strongly suggest to recruit more healthy participants and re-analyzed again.

As you indicated, the sample size of the HC group was small. It would have been ideal if the number of HCs would be the same as the patients, but it was difficult to recruit more controls because we could not find more new controls until the study’s approval had expired. However, the results were reliable for the number because the population of the controls was homogeneous in terms of the SDs and interquartile range despite the controls being recruited randomly. We believe that the results of this study are significant even with 10 people in the HC group. Of course, the confirmation study should be conducted in another study protocol. We added “Third, the number of HCs was small. It would have been ideal if the number of HCs was the same as the number of patients, but it was difficult since we could not recruit more controls. This was due to our inability to find new HC’s as the study’s approval had also expired. However, the results were reliable for the number chosen owing to the homogeneity of the control group in terms of the SD’s and interquartile range despite the controls being recruited randomly. A confirmation study to verify the same could be conducted at a later point in time. Line 542

> 2. In Introduction section, more special description about the different evoked potential, what specific evoked potential reflect physiological feature? It is wonderful if hypothesis was present based on physiological feature.

As you indicated, the Introduction section was insufficient to describe the associations between evoked potential and physiological feature. Based on previous studies, we added associations between VEP P100 latency and polysynaptic visual pathway, SEP N30 amplitude and a cortico-subcortico-cortical reentry loop, and ABR wave V latency and brainstem dysfunction. Therefore, we added “Till date, it is considered that dopamine deficiency affects polysynaptic visual pathway, and an increase in the latency of the P100 component of the visual/visually evoked potential (VEP) is reported in PD [2]. In addition, a decrease in the amplitude of the N30 component of the somatosensory evoked potential (SEP) is reported in PD [3]. This could be since the frontal P20-N30-P40 complex (responsible for SEP) is generated with a cortico-subcortico-cortical reentry loop indicating cortical dysfunction in PD. An increase in the latency of wave V (seen as AEP) of the auditory brainstem response has also been reported in PD [4], and it is known that brainstem dysfunction is present in PD patients with dementia. Line 63

> 3. In Clinical characteristics and measures, cognitive function should be provided for all tasks and its sub-domains.

As we adopted as much as 10 cognitive tests, the clinical characteristics and measures section has become very complicated. In addition, some of them are not familiar to people other than Japanese. To provide cognitive function for all tasks and its sub-domains, we added: 

These are accordingly listed in Table 1.

Note that, each task for attention is designed to evaluate a specific attention, but it is also affected by other attention such as working memory, sustained attention and divided attention. Line 103.

> 4. There are patients in the off-state and on-state, what difference between two kinds of patients.

We are sorry for lacking the description about the patients in the off- state and on-state. We tested UPDRS part 3 in the on-state on all 20 patients, and 8 of the 20 patients were also tested UPDRS part 3 in the off-state: The 8 patients were inpatients to assess DBS adaptation. We added “eight patients out of the total 20 patients were administered UPDRS part 3 in the off-state since their antiparkinsonian drugs were withdrawn for the purpose of assessing deep brain stimulation (DBS) indications. Line 184

> 5. Multiple corrections should also be conducted for group comparison.

As you pointed out, we did not perform Bonferroni correction in group comparisons, and statistical analysis methods were not unified. 

For cognitive tests, VCT time of simple symbol, complex symbol and number and ADT hit rate of the CAT remained significant after Bonferroni correction. From this result, sustained attention deficits are likely to be in PD because VCT and ADT remained significant. We added “Especially, VCT time of simple symbol, complex symbol and number and ADT hit rate of the CAT remained significant after Bonferroni correction.” on line 188 as a result, and “

“Sustained attention deficits are likely to be present in PD since among all the cognitive tests performed in our study, VCT and ADT remained significant after Bonferroni correction. Line 285.

For evoked fields, SEF P60m (R, M), P60m – N20m (R, M) and P60m - P35m (R, M) remained significant after Bonferroni correction. As SEF P60m (R, M) (and SEF P60m – P35m (R, M) and SEF P60m – N20m (R, M)) remained significant after Bonferroni correction among all the evoked fields, it is strongly suspected that P60m is affected by PD pathology. We added “Especially, SEF P60m (R, M), P60m – N20m (R, M) and P60m - P35m (R, M) remained significant after Bonferroni correction.” on line 223 as a result, and

“However, as SEF P60m (R, M), SEF P60m – P35m (R, M) and SEF P60m – N20m (R, M) values remained significant after Bonferroni correction among all the evoked fields, it is strongly suspected that P60m is affected by underlying PD pathology.” Line 471

In addition, for site location, there were no significant differences between patients with PD and HCs after Bonferroni correction. We added “Further, after applying Bonferroni correction the values for these three components became insignificant. Line 234.

> 6. In Conclusion section, authors stated that evoked fields reflect basal ganglia function, what cognitive function or evoked fields are linked with ganglia function based on your results or previous studies?

As you pointed out, Conclusions section was just a list of Results section. We added 

“We investigated the characteristics of cognitive functioning and evoked fields in patients with PD. PD patients presented with cognitive impairment on assessment tests such as MoCA-J, S-PA, FAB, TMT-A, B, CAT, ADT and VPTA. Especially, sustained attention deficits are likely to be present in PD. The latencies of the VEF N75m, P100m, N145m, AEF P50m, P100m, and SEF P60m components were greater in PD patients than in HCs. The increased latencies of VEF, AEF, and SEF were correlated mainly with medication and motor symptoms and less with cognitive tasks. Interestingly, AEF P100m and SEF P60m latency could strongly reflect basal ganglia functioning as per our findings. These specific responses are candidates for assessing motor symptoms or drug treatment effects in PD patients.” Lines 553-561.

---

## [Decision Letter · Decision Letter 1]

1 Sep 2020

Clinical factors affecting evoked magnetic fields in patients with Parkinson's disease

PONE-D-20-11314R1

Dear Dr. Yabe,

We’re pleased to inform you that your manuscript has been judged scientifically suitable for publication and will be formally accepted for publication once it meets all outstanding technical requirements.

Kind regards,

Wing-ho Yung, PhD

Academic Editor

PLOS ONE

Additional Editor Comments (optional):

Reviewers' comments:

Reviewer's Responses to Questions

**Comments to the Author**

1. If the authors have adequately addressed your comments raised in a previous round of review and you feel that this manuscript is now acceptable for publication, you may indicate that here to bypass the “Comments to the Author” section, enter your conflict of interest statement in the “Confidential to Editor” section, and submit your "Accept" recommendation.

Reviewer #2: All comments have been addressed

2. Is the manuscript technically sound, and do the data support the conclusions?

Reviewer #2: Yes

3. Has the statistical analysis been performed appropriately and rigorously? 

Reviewer #2: Yes

4. Have the authors made all data underlying the findings in their manuscript fully available?

Reviewer #2: Yes

5. Is the manuscript presented in an intelligible fashion and written in standard English?

Reviewer #2: Yes

6. Review Comments to the Author

Reviewer #2: (No Response)

7. PLOS authors have the option to publish the peer review history of their article (what does this mean?). If published, this will include your full peer review and any attached files.

Reviewer #2: **Yes: **Yuanchao Zhang

---

## [Editor Report · Acceptance letter]

8 Sep 2020

PONE-D-20-11314R1 

Clinical factors affecting evoked magnetic fields in patients with Parkinson's disease 

Dear Dr. Yabe:

I'm pleased to inform you that your manuscript has been deemed suitable for publication in PLOS ONE. Congratulations! Your manuscript is now with our production department. 

Kind regards, 

on behalf of

Dr. Wing-ho Yung 

Academic Editor

PLOS ONE